# In-distribution adversarial attacks on object recognition models using gradient-free search.

**Spandan Madan**                                                    *spandan.madan@gmail.com*
*Harvard University*

**Tomotake Sasaki**                                                  *tomotake.sasaki@fujitsu.com*
*Fujitsu Limited**

**Hanspeter Pfister**                                                *pfister@seas.harvard.edu*
*Harvard University*

**Tzu-Mao Li**                                                       *tzli@ucsd.edu*
*UCSD*

**Xavier Boix**                                                      *xboix@fujitsu.com*
*Fujitsu Research of America*

**Reviewed on OpenReview:** `https://openreview.net/forum?id=uF9ZdAwrCT`

## Abstract

Neural networks are susceptible to small perturbations in the form of 2D rotations and shifts, image crops, and even changes in object colors. Past works attribute these errors to dataset bias, claiming that models fail on these perturbed samples as they do not belong to the training data distribution. Here, we challenge this claim and present evidence of the widespread existence of perturbed images within the training data distribution, which networks fail to classify. We train models on data sampled from parametric distributions, then search *inside* this data distribution to find such in-distribution adversarial examples. This is done using our gradient-free evolution strategies (ES) based approach which we call CMA-Search. Despite training with a large-scale ($\sim 0.5$ million images), unbiased dataset of camera and light variations, CMA-Search can find a failure inside the data distribution in over 71% cases by perturbing the camera position. With lighting changes, CMA-Search finds misclassifications in 42% cases. These findings also extend to natural images from ImageNet and Co3D datasets. This phenomenon of in-distribution images presents a highly worrisome problem for artificial intelligence—they bypass the need for a malicious agent to add engineered noise to induce an adversarial attack. All code, datasets, and demos are available at https://github.com/Spandan-Madan/in_distribution_adversarial_examples.

## 1 Introduction

Neural networks are highly susceptible to small perturbations—2D rotations and translations Engstrom et al. (2018), image crops Srivastava et al. (2019); Azulay & Weiss (2019), and even changes in the color space Mohapatra et al. (2020); Hosseini & Poovendran (2018); Shamsabadi et al. (2020). Existing works have claimed that these failures lie out of the training data distribution, and attribute these failures to dataset bias Ilyas et al. (2019); Lee et al. (2018); Grosse et al. (2017); Karunanayake et al. (2024); Stutz et al. (2019). Here, we put this hypothesis to test by training classification models on datasets with explicitly controlled train/test distributions, and searching for adversarial examples *within* the training data distribution.

---

*Tomotake Sasaki is currently affiliated with Tokyo Metropolitan, Chuo-Johoku Vocational Skills Development Center / Japan Electronics College (current email address: 24eo0111@jec.ac.jp).

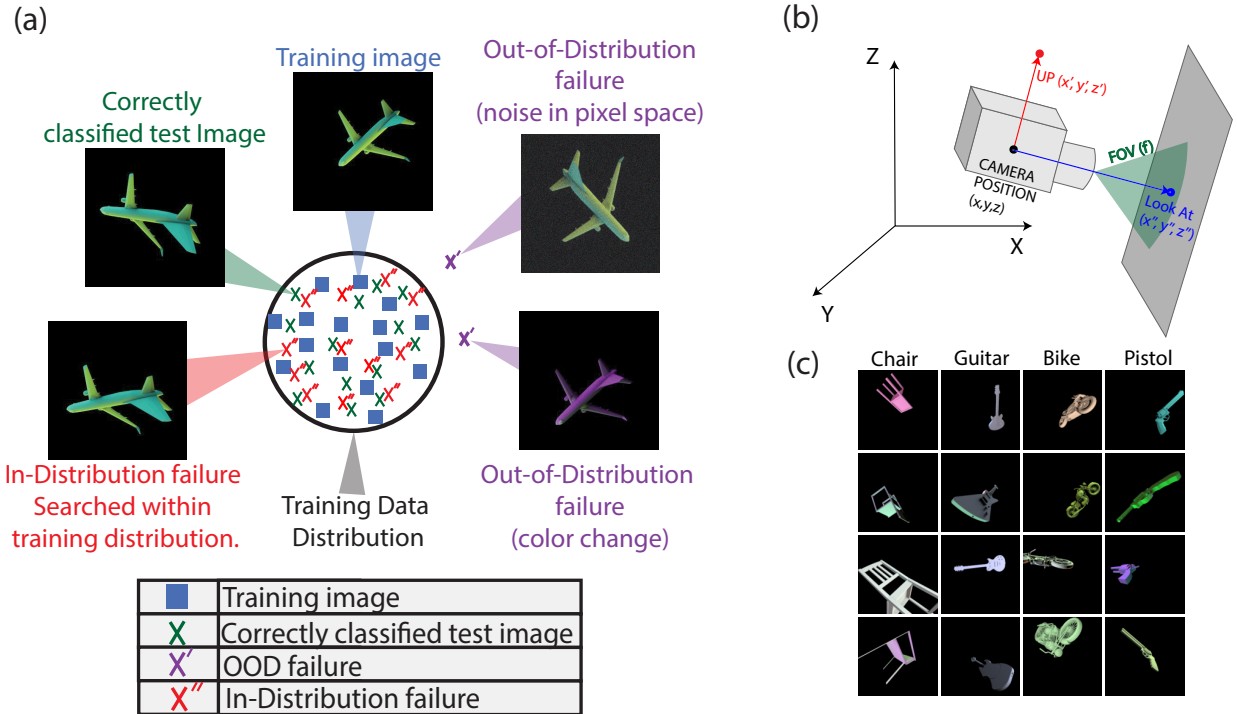

Figure 1: *In-distribution adversarial attacks.* (a) The data distribution (depicted in black) refers to the space of all camera and light variations. Typical adversarial examples are created by adding noise to the image, which may result in images out of the data distribution. CMA-Search finds failures inside the data distribution. (b) 3D scene setup for our rendered images with camera parameters illustrated. (c) Example images with camera and light variations.

Our key finding is that there is a widespread presence of adversarial examples within the training distribution, as illustrated in Fig. 1(a). Thus, networks are highly susceptible to small perturbations not just out of the training distribution as previously known, but inside the training distribution as well. In practice, these in-distribution adversarial examples point to a highly worrisome problem—these failures bypass the need for a malicious agent to induce an error. These experiments are enabled by our gradient-free, evolutionary strategies (ES) based approach for finding in-distribution adversarial examples, which we call CMA-Search.

We present results with CMA-Search across three levels of data complexity—(i) parametric data sampled from disjoint per-category uniform distributions, (ii) parametric and controlled data of rendered images, and (iii) natural image data from ImageNet and Co3D datasets.

Across all datasets, models are highly susceptible to in-distribution adversarial attacks. CMA-Search can find in-distribution attacks for simplistic parametric data with a 100% attack rate—there existed a failure in the vicinity of every single correctly classified test point. For rendered data, CMA-Search found failures in the vicinity of 71% correctly classified images by perturbing the camera position, and for 42% images by perturbing lighting parameters. With natural images from the Common Objects in 3D (Co3D) dataset Reizenstein et al. (2021), CMA-Search found in-distribution adversarial examples for over 51% images. Finally, we also employed CMA-Search in conjunction with a novel view synthesis pipeline Tucker & Snavely (2020) to find in-distribution adversarial examples in the vicinity of ImageNet Deng et al. (2009).

## 2 Related Work

In efforts to combat susceptibility to small transformations Engstrom et al. (2018), crops Srivastava et al. (2019); Azulay & Weiss (2019), and 2D rotations and translations Alcorn et al. (2019), alternative architectures

have been proposed which are shift invariant. This includes anti-aliasing networks using the seminal signal processing trick of anti-aliasing Zhang (2019), and recently proposed truly shift invariant networks which use a new sampling methodology to guarantee a 100% consistency in classification under 2D shifts Chaman & Dokmanić (2021). Unlike our work, these works have focused only on 2D transformations.

Recent work has also sought to generate adversarial perturbations which are human interpretable i.e. semantic adversarial examples. These works often rely on synthetic data, using differentiable rendering or other optimization methods to find adversarial images by modifying scene parameters Liu et al. (2019); Zeng et al. (2019); Shetty et al. (2020); Jain et al. (2019); Xiao et al. (2019); Joshi et al. (2019); Yao et al. (2020). These include a custom differentiable renderer to perturb the camera, lighting, or object mesh vertices, and using a neural renderer where light is represented by network activations.

They key differences between these works and ours is that our adversarial attacks are guaranteed to lie within the training distribution. While in-distribution attacks have been shown in theoretical works and for toy data Gilmer et al. (2018); Fawzi et al. (2016; 2018a;b), this work provides the first evidence of such failures with real-world data to the best of our knowledge.

## 3    Datasets with explicitly controlled data distributions

Mathematically, a sample $x^*$ is defined to be in-distribution w.r.t. a dataset $X = \{x_1, ...x_N\}$, if $x^*$ and all points $(x_1...x_N)$ are generated by sampling i.i.d. from the same generative distribution. Thus, as $n \to \infty$, $x^* \in X$. As an example, consider the Camera Position. Our dataset with camera and lighting variations was constructed by sampling rendering images with camera position uniformly sampled from $[0.5, 8]$ units. Thus, any image of a scene with camera position within the range $[0.5, 8]$ is in-distribution. Images with camera position not in this range are considered out of distribution. For all datasets, we sample uniformly across the support. This choice allows the support to uniquely characterize the data distribution.

### 3.1    Generating simplistic parametrically controlled data

We created a binary classification task by sampling data from two $N$-dimensional uniform distributions confined to disjoint ranges $(a, b)$ and $(c, d)$, as described in the following:

$$x_i \sim \left\{ \begin{array}{ll} \text{Unif}(a, b, N); & y_i = 0 \\ \text{Unif}(c, d, N); & y_i = 1 \end{array} \right\}. \tag{1}$$

We set $a = -10, b = 10, c = 20, d = 40$ for experiments presented. However, we observed that the exact choice of these parameters does not impact the findings. To generate an in-distribution test set, we simply sample new data points from the training distribution. This procedure is consistent with recent theoretical work on the adversarial attacks Gilmer et al. (2018); Fawzi et al. (2018a).

### 3.2    Generating an unbiased training dataset of camera and light variations

Large-scale datasets for computer vision have mostly been created by scraping pictures from the internet Deng et al. (2009); Lin et al. (2014); Everingham et al.; Krause et al. (2013); Zhou et al. (2017). However, investigating in-distribution robustness requires sampling new points from regions of interest within the data distribution, which is not possible with these datasets. To address this issue, we use a computer graphics pipeline for generating and modifying images which ensures complete parametric control over the data distribution. We simply sample camera and lighting parameters from a fixed, uniform distribution, and render a subset of 3D models from ShapeNet Chang et al. (2015) objects with the sampled camera and lighting parameters.

All models were trained on 0.5 million rendered images across 11 categories, with 1000 images for every 3D model. Each image was constructed by rendering a frame from the 3D scene setup illustrated in Fig. 1(b)— one camera, one 3D model and 1-4 lights. Thus, every image is parametrized by the camera and the light parameters. The camera parameters are 10 Dimensional, and each light is 11 dimensional. Multiple lights ensure that scenes contain complex mixed lighting, including self-shadows. There is a one-to-one mapping

between the pixel space (rendered images) and the $(11n + 10)$ camera and light parameters, with $n =$ number of lights. Sample images are shown in Fig. 1(c) and Fig. S3. All camera and light parameters were sampled from uniform distributions with pre-specific ranges described in the supplement. For these parameters, and additional details please refer to Sec. S1.

### 3.3 Natural image datasets—ImageNet and Common Objects in 3D

As a real litmus test, we also ensure that our findings hold true for natural images. We present results on two popular natural image datasets—ImageNet Russakovsky et al. (2015) and the Common Objects in 3D (Co3D) Reizenstein et al. (2021) dataset.

**Co3D:** This dataset was originally created by users capturing short videos of fixed objects placed on a surface by a user moving a mobile phone around the object with adjacent frames representing nearby 3D views. We utilize this to test in-distribution robustness. The training dataset was constructed by sampling uniformly across videos from 5 categories (car, chair, handbag, laptop, and teddy bear). This amounts to $187,200$ training images, or $38,000$ images per category which is 32 times the ImageNet training set on a per category basis. An in-distribution test set of $68,854$ images was generated by sampling the remaining frames from these categories. Thus, the test set represents interpolated viewpoints between training viewpoints. Once models are trained, our approach searched within 5 adjacent frames to find an in-distribution failure. Additional details can be found in Sec. S2.2.

**ImageNet:** A Novel View Synthesis (NVS) Tucker & Snavely (2020) model was used to generate views in the vicinity of images. (See Sec. S2.1 for details). The NVS model takes as input an image and the $(x, y, z)$ offsets which describe camera movement along the X, Y and Z axes. Unlike our renderer, it cannot introduce changes to the camera Look At, Up Vector, Field of View or lighting changes. CMA-Search optimizes these offset parameters of the NVS model to find a perturbed image which is misclassified.

## 4 CMA-Search: Finding in-distribution failures by searching the vicinity

CMA-Search can be used to attack any parametric dataset. The methodology starts with the parametric representation of a correctly classified input, and optimizes these parameters using Covariance Matrix Adaptation-Evolution Strategy (CMA-ES) Hansen & Ostermeier (1996); Hansen (2016) to find a misclassified sample in the vicinity of the start point. Algorithm 1 provides an outline for the method which was implemented using pycma Hansen & Ostermeier (1996); Hansen et al. (2019). We explain the methodology with an example of finding in-distribution adversarial attacks within the distribution of camera parameters. The algorithm for searching adversarial attacks in the space of light parameters, and for attacking all other datasets is analogous. For ease, the approach is also visualized as a flowchart in Fig. S1

Starting from the initial camera parameters of the scene, CMA-ES generates offspring by sampling from a multivariate normal (MVN) distribution i.e. mutating the original parameters. These offspring are sorted based on the fitness function $(1 - p,$ where $p$ denotes classification probability). The best offspring are used to modify the mean and covariance matrix of the MVN for the next generation. The mean represents the current best estimate of the solution i.e. the maximum likelihood solution, while the covariance matrix dictates the direction in which the population should be directed in the next generation. The search is stopped either when a misclassification occurs, or after 15 iterations. At each generation, 10 offspring were generated. For results presented on the simplistic parametrically controlled data, we checked for a misclassification till 1500 iterations and 20 offspring were generated in each iteration. CMA-ES is an unconstrained optimization procedure. Thus, we ensured that a misclassification counts as an in-distribution adversarial attack only if it met both criteria—(1) it caused a misclassification, (2) it belongs to the training data distribution (*i.e.,* inside the support of the underlying distribution).

**Evaluating CMA-Search:** For all models, we report the Attack Rate—the percentage of correctly classified points for which CMA-Search successfully found a misclassification. A model with no in-distribution failures would have an attack rate of 0, making this a natural goal for benchmarking studies using this metric.

---

**Algorithm 1** *CMA-Search* over camera parameters to find in-distribution adversarial examples.

---

Let $x \in \mathbb{R}^{10}$ denote the camera parameters.
Let *Render* and *Network* denote the rendering pipeline and classification network respectively.
**function** FITNESS($x$, *Render*, *Network*)
    image $= Render(x)$
    predicted_category, probability $= Network$(image)
    **return** predicted_category, probability
**end function**

$x_{init}$: initial camera parameters, $\lambda$: number of offspring per generation (set to 20), $y$: image category, and $R_T$: Range of camera parameters in training data.
**procedure** CMA-SEARCH($x_{init}, \lambda, y, D$)
    **initialize** $\mu = x_{init}, C = I$                        ▷ *I* denotes identity matrix.
    iters $= 0$
    **while** iters $< 20$ **do**
        **for** $j = 1, ..., \lambda$ **do**
            $x_j =$ sample_multivariate_normal($\mu, C$)          ▷ Generate mutated offspring
            $y_j, p_j =$ FITNESS($x_j$, *Render*, *Network*)          ▷ Calculate fitness of offspring
            **if** $y_j \neq y$ **then**          ▷ Classification fails for image with camera parameters $x_j$
                **if** $x_j \in R_T$ **then**          ▷ $x_j$ is in in-Distribution failure
                    **return** True
                **end if**
            **end if**
        **end for**
        $x_{1...\lambda} \leftarrow x_{s(1)...s(\lambda)}$, with $s(j) = \text{argsort}(p_j)$          ▷ Pick best offspring
        $\mu, C \leftarrow$ update_parameters($x_{1...\lambda}, \mu, C$)
        iters $=$ iters $+1$
    **end while**
    return False
**end procedure**

---

For simplistic parametrically controlled data, the *Attack Rate* was measured by attacking $20,000$ correctly classified samples. Due to our use of a physically based renderer that accurately models the physics of light in the 3D scene, generating images in the vicinity of the correctly classified image is a computational intensive process. For rendered data, it was measured by attacking $2,000$ correctly classified images for every architecture. For one model (ResNet18) we also measured the *Attack Rate* with $20,000$ images as an additional control. For the Co3D dataset, it was measured on $116,850$ images.

**Visualizing the vicinity of an error:** CMA-Search generates an in-distribution error $(\vec{x_a})$ starting from a correctly classified point $(\vec{x_c})$. Using these two points, we defined a unit vector in the adversarial direction and set it as one basis vector $(\vec{e_1} = \vec{x_a} - \vec{x_c})$. For $D$ dimensional data, we computed the remaining $D-1$ orthonormal bases, and randomly selected one as the orthogonal direction $(\vec{e_2}, \text{ such that } \vec{e_2} \perp \vec{e_1})$. Following past work Warde-Farley & Goodfellow (2016), we defined a grid of perturbations along the adversarial and the picked orthogonal direction. The classification model was then evaluated on noisy samples constructed by adding noise on the grid formed by these two basis vectors:

$$\vec{x_i} = \vec{x_c} + \alpha \vec{e_1} + \beta \vec{e_2} \tag{2}$$

Here, $\alpha, \beta \in [0,1]$ with intervals of 0.01. The church-window plot shows classification on these perturbed samples, with correct classifications shown in white, in-distribution adversarial examples in red, and out-of-distribution samples in black.

## 5 Experimental Details

Below we provide the training details including model architectures, optimization strategies and other hyper-parameters used for the binary classification models trained on simplistic parametrically controlled data, and the object recognition models trained on our rendered images of camera and light variations. All code to run these experiments can be found at https://github.com/in-dist-adversarials/in_distribution_adversarial_examples.

### 5.1 Training details for MLPs for classifying parametrically controlled uniform data

Let $D$ denote data dimensionality, and $N$ denote dataset size. A 5 layer multi-layer perceptron (MLP) with ReLU activations was used, with the output dimensionality of hidden layers set to $5D$, $D$, $D/5$, $D/5$, and 2 respectively. However, we found that the number of MLP hidden layers and the number of neurons in these layers had no significant impact on trends of in-distribution robustness. For experiments with $N < 64,000$ all data was passed in a single batch. For experiments with more data points, each batch contained $64,000$ points. All models were trained for 100 epochs with stochastic gradient descent (SGD) with a learning rate of 0.0001. All experiments were conducted on a compute cluster consisting of 8 NVIDIA TeslaK80 GPUs, and all models were trained on a single GPU at a time. Only models achieving a near perfect accuracy $(> 0.99)$[1] on a held-out test set were attacked using *CMA-Search*.

### 5.2 Training details for Object recognition models for classifying images of real-world objects

All CNN models were trained with a batch size of 75 images, while transformers were trained with a batch size of 25. Models were trained for 50 epochs with an Adam optimizer with a fixed learning rate of 0.0003. Other learning rates including 0.0001, 0.001, 0.01 and 0.1 were tried but they performed either similarly well or worse. To get good generalization to unseen 3D models and stable learning, each image was normalized to zero mean and unit standard deviation. As before, all experiments were conducted on our cluster with TeslaK80 GPUs, and each model was trained using a single GPU at a time.

---

[1]Except when dataset size=1000 and dimensions=100 or 500. In these two case the training data was too small for a high test accuracy. These cases are still included for completion.

# 6 Results

We report results on in-distribution adversarial attacks on classification models trained across four datasets—(i) simplistic data sampled from disjoint per-category uniform distributions (Sec. 6.1), (ii) parametrically controlled images of objects using our graphics pipeline (Sec. 6.2), (iii) Common Objects in 3D dataset Reizenstein et al. (2021)(Sec. 6.3), and (iv) ImageNet Deng et al. (2009)(Sec. 6.3) For each model, we report the attack rate—the percentage of correctly classified points for which we successfully found an in-distribution failure using CMA-Search. Additional details on the implementation and evaluation of CMA-Search are reported in Sec.S3.

## 6.1 In-distribution adversarial attacks on uniformly distributed data

Fig. 2(a) reports the attack rate for models—the percentage of correctly classified points for which we successfully found an in-distribution failure using CMA-Search. Despite a near perfect accuracy on a held-out test set, in-distribution adversarial examples can be identified in the vicinity of all correctly classified test points—the attack rate is 100% for models trained with 20, 100 and 500 dimensional data. Note that this simplistic dataset is easily separable by the simplest of models including a decision tree. However, DNNs trained on this dataset are plagued by in-distribution failures.

**Impact of dataset size:** The attack rate start dips once a critical dataset size is reached (Fig. 2(a)). However, data complexity scales poorly with number of dimensions. As dimensionality grows from 20 to 100, the number of points required for robustness scales almost 100-fold. For 500 dimensions even 10 million training points were not sufficient. On average, only 51 iterations were needed to find a misclassification for 10 dimensional data. This dropped to 20 iterations for 100 dimensional data and 11 for 500 dimensional data.

**Impact of robust training:** We fine-tuned models on 20,000 in-distribution adversarial examples found using CMA-Search for 100 dimensional data. The attack rate stayed at 100%, with no improvement in model robustness against CMA-Search. This is expected, as our identified adversarial examples lie within the training distribution. Thus, robust training in this case essentially amounts to a marginal increase in the training dataset size which is already discussed above.

Fig. 2(b) reports the average distance between the (correctly classified) start point and the closest in-distribution adversarial example identified using CMA-Search. This distance increased with dataset size. At critical dataset sizes, adversarial examples are far enough from starting points that they are now not in-distribution. This results in the dip in the attack rate shown in Fig. 2(a).

**Visualizing failures:** Fig. 2(c) shows the learned decision boundary using church window plots Warde-Farley & Goodfellow (2016) (see S3.3 for details). Intriguingly, there is a clean transition from correctly classified points (white) to in-distribution adversarial examples near the decision boundary (red), beyond which points become out of the distribution (black). Thus, in-distribution adversarial examples are isolated to a region close to the category boundary, and in a contiguous fashion. This finding has been theorized Fawzi et al. (2016; 2018a); Gilmer et al. (2018); Fawzi et al. (2018b), but to the best of our knowledge this is the first empirical evidence for this phenomenon.

## 6.2 Networks struggle to generalize across camera and light variations

Here, we present the first evidence of in-distribution adversarial attacks on visual recognition models. Despite 0.5 million images for 11 categories with over 1000 images for every $3D$ model, CMA-Search found small changes in 3D perspective and lighting which had a catastrophic impact on network performance, as shown in Fig. 3(a).

Table 1 reports in-distribution adversarial attacks identified by CMA-Search using small changes in 3D perspective and lighting. For 71% images correctly classified by a ResNet, there lies an in-distribution failure within a 1.83% change in the camera position. For transformers, the impact is far worse with an Attack

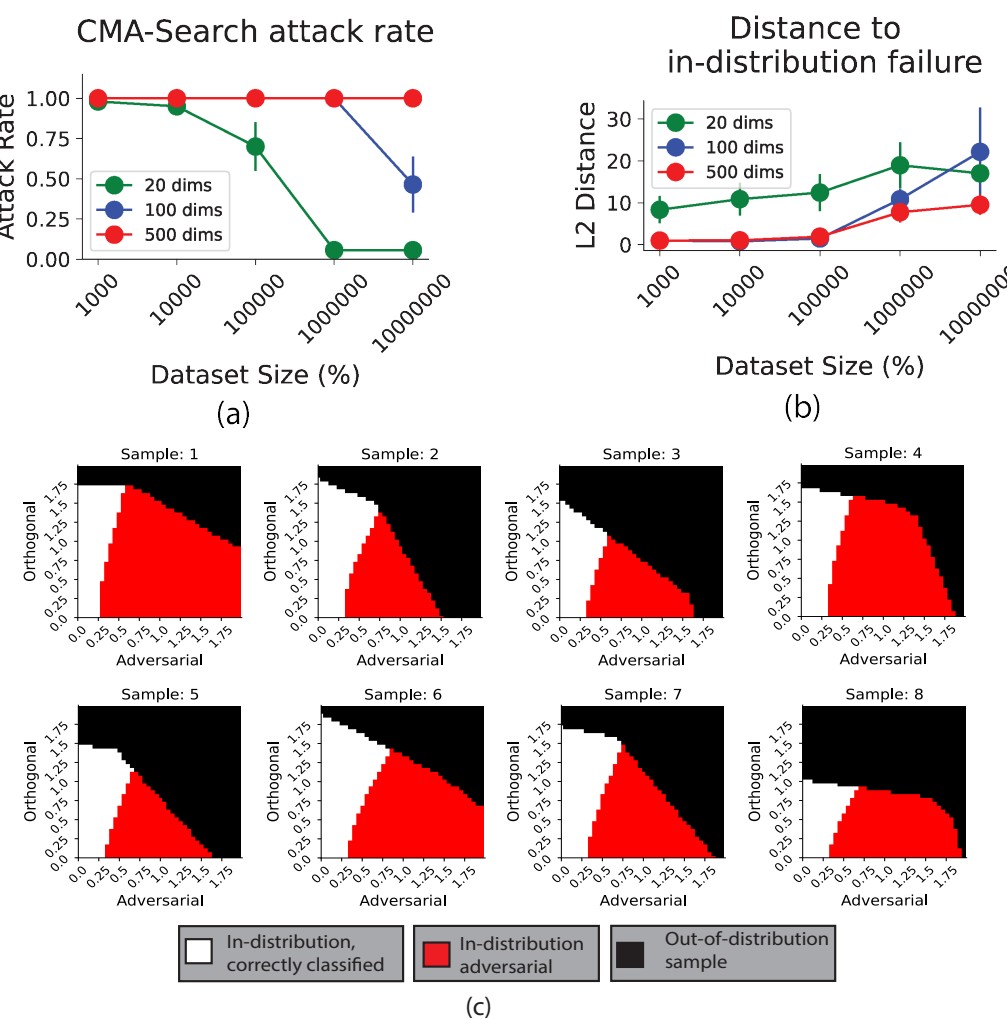

Figure 2: *In-distribution adversarial attacks on parametric data sampled from high-dimensional, disjoint uniform distributions.* (a) Attack rate measured using CMA-Search is 100% for all models—there exists an in-distribution failure in the vicinity of every correctly classified sample. Models become robust beyond a critical dataset size, but the data needed scales poorly with dimensionality. (b) Average Euclidean distance between the starting point and the identified in-distribution adversarial sample increases as dataset size increases. (c) Church window plots depicting adversarial examples (red) located contiguously and in between the learned and ground-truth boundaries.

Rate of 85%. For lighting changes, CMA-Search can find a misclassification in 42% cases with just a 6.5% change. On average, 2 iterations were needed to find an in-distribution failure with camera variations. For light variations, 3.5 iterations were required on average.

All architectures were most sensitive to changes in the Camera Position and the Camera Look At—subtle, in-distribution 3D perspective changes. Shift-invariant architectures designed to improve robustness to 2D shifts performed better, they were still highly susceptible to 3D perspective changes (see Table 1).

Fig. 3(b) shows the distribution of scene parameters for misclassified images. Errors are distributed across the space with no clear, strong patterns characterizing the camera and light conditions where networks struggle. This is in stark contrast to human vision, which is well-documented to be significantly impacted by changes in camera parameters in the form of canonical vs. non-canonical poses Gomez et al. (2008); Terhune et al. (2005); Blanz et al. (1999), and upside-down orientationsKöhler (1960); Lewis (2001); Thompson (1980),

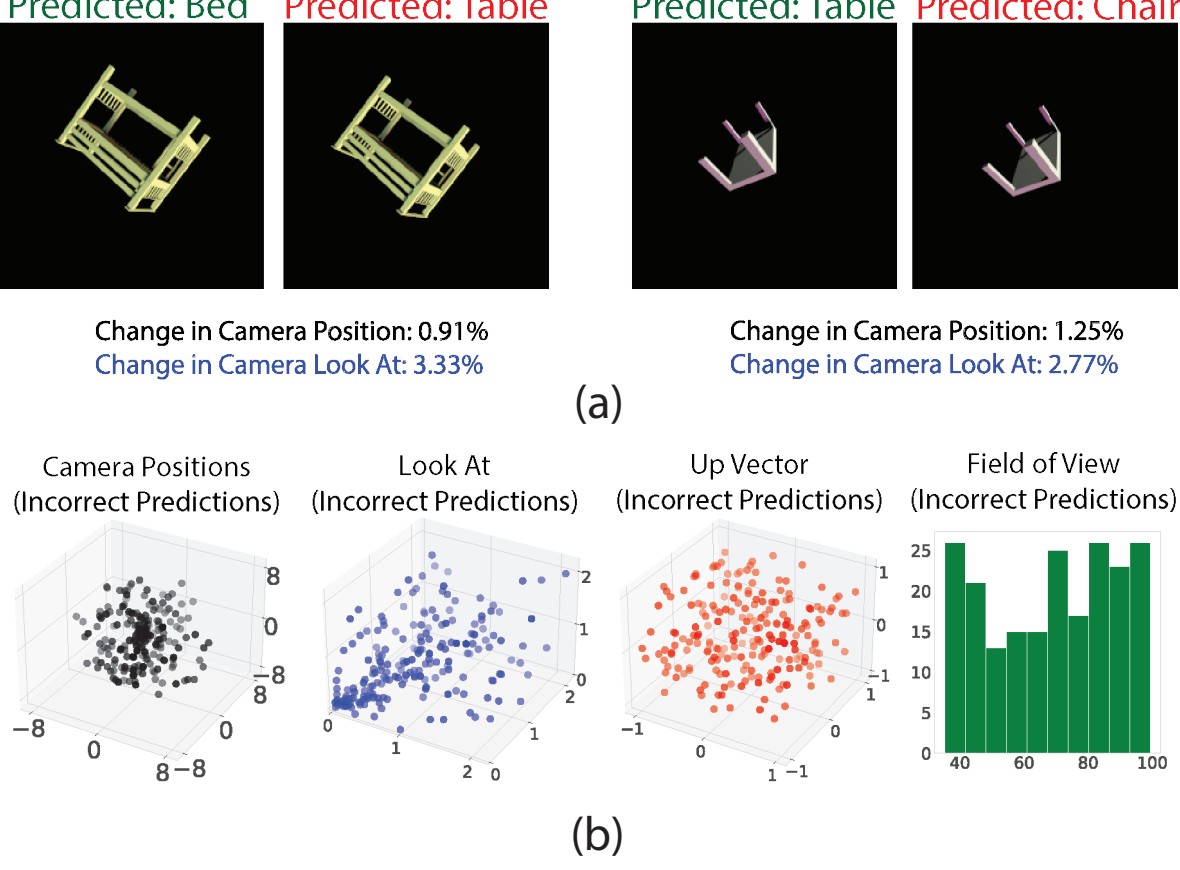

Figure 3: *In-distribution adversarial attacks in the camera parameter space.* a) Sample in-distribution adversarial examples. Percentage of change in Camera Position and Camera Look At parameters needed to induce the misclassification are also reported. Attack rates are reported in Table 1. (b) Distribution of camera parameters for in-distribution adversarial images. Unlike human vision, there were no clear patterns characterizing the camera and light conditions of misclassified images.

| Model Architecture | CMA Cam | | CMA Light | |
|---|---|---|---|---|
| | Attack Rate (%) | Distance (mean ± std) | Attack Rate (%) | Distance (mean ± std) |
| ResNet18 | 71 | 1.83 ± 1.33 | 42 | 6.52 ± 5.68 |
| Anti-Aliased Networks | 45 | 2.32 ± 2.09 | 40 | 7.03 ± 5.10 |
| Truly Shift Invariant Network | 53 | 2.22 ± 2.16 | 25 | 6.72 ± 5.41 |
| ViT | 85 | 1.34 ± 1.16 | 65 | 4.63 ± 3.49 |
| DeIT | 85 | 1.27 ± 0.81 | 51 | 4.54 ± 2.75 |
| DeIT Distilled | 86 | 1.22 ± 0.87 | 55 | 4.49 ± 2.27 |

Table 1: *Attack Rates for models attacked with CMA-Search over camera and light parameters.* CMA-Search starts with correctly classified images, and searches the space of camera and light parameters to find an in-distribution misclassification. The attack rate reports percentage of correctly classified images for which CMA-Search found a failure. The change in parameter space (mean distance) required to induce an error is extremely small, highlighting the brittleness of these models.

among others. In the supplement we provide additional results reporting CMA-Search over camera and light parameter space (See Sec. S4).

Table 2: *Results with Co3D dataset.* All models suffer from high attack rates, confirming the widespread presence of in-distribution failures for object recognition models.

|  | ResNet | Anti-Aliased Networks | ViT | DeIT |
|---|---|---|---|---|
| **Test Accuracy** | 0.92 | 0.94 | 0.82 | 0.85 |
| **Attack Rate** | 0.51 | 0.39 | 0.72 | 0.72 |

Combined, these results confirm that object recognition models are plagued by in-distribution adversarial attacks.

### 6.3 Results on Natural Image Data

**Results on Co3D:** Table 2 reports the average accuracy and attack rate for models trained on Co3D. Despite a high test accuracy of 92%, a ResNet model suffered from an attack rate of 51%. Thus, there were in-distribution adversarial examples within 1-5 frames of the correctly classified frame for over half the images. Sample failures are provided in Fig. 4(a) Transformers struggled even more, with ViT and DeIT Touvron et al. (2021) having an attack rate near 72%. The shift invariant architecture was more robust, but attack rate was still high at 39% (see Table 2). These trends are consistent with the results in Table 1.

**Results on ImageNet:** We also confirmed that these results extend to ImageNet. We present empirical results for a ResNet18 model trained on ImageNet, and OpenAI's transformer-based CLIP model Radford et al. (2021) in Fig. 4(b). Additional ImageNet failures found using CMA-Search are provided in Fig. S5. Furthermore, Fig. S6 presents results analyzing how the noise introduced by the NVS pipeline impacts classification performance in finer-grained detail.

Combined, these results across 4 datasets confirm that despite near-perfect test set accuracies and millions of training examples, classification models struggle with the widespread presence of in-distribution adversarial examples.

## 7 Conclusions

Susceptibilities of recognition models have often been attributed to biased training data. Here, we put this hypothesis to test by training and testing with a large-scale, unbiased dataset and propose a new search method for investigating the brittleness of neural networks, which we call CMA-Search. We conducted experiments with 4 datasets ranging from simple parametric data, to a rendered dataset of camera and light variations, and finally natural image datasets ImageNet and Co3D.

Across all datasets our findings are consistent—while data augmentation, unbiased datasets, and specialized shift-invariant architectures are certainly helpful in improving model robustness, the real problem runs far deeper. Despite high test accuracies, networks are plagued by adversarial examples that lie within the training distribution as measured by the attack rate.

## 8 Discussions

In practice, these in-distribution adversarial examples point to a highly worrisome problem—these failures bypass the need for a malicious agent to induce an error. Even when the model has a near perfect test accuracy, these examples lie hidden within the data distribution in plain sight.

This discovery of in-distribution adversarial examples presents critical security challenges distinct from traditional adversarial attacks. While conventional attacks require engineered perturbations, these adversarial examples exist naturally within the expected data distribution, creating two severe vulnerabilities. First, they bypass traditional detection methods that rely on identifying out-of-distribution characteristics or unusual

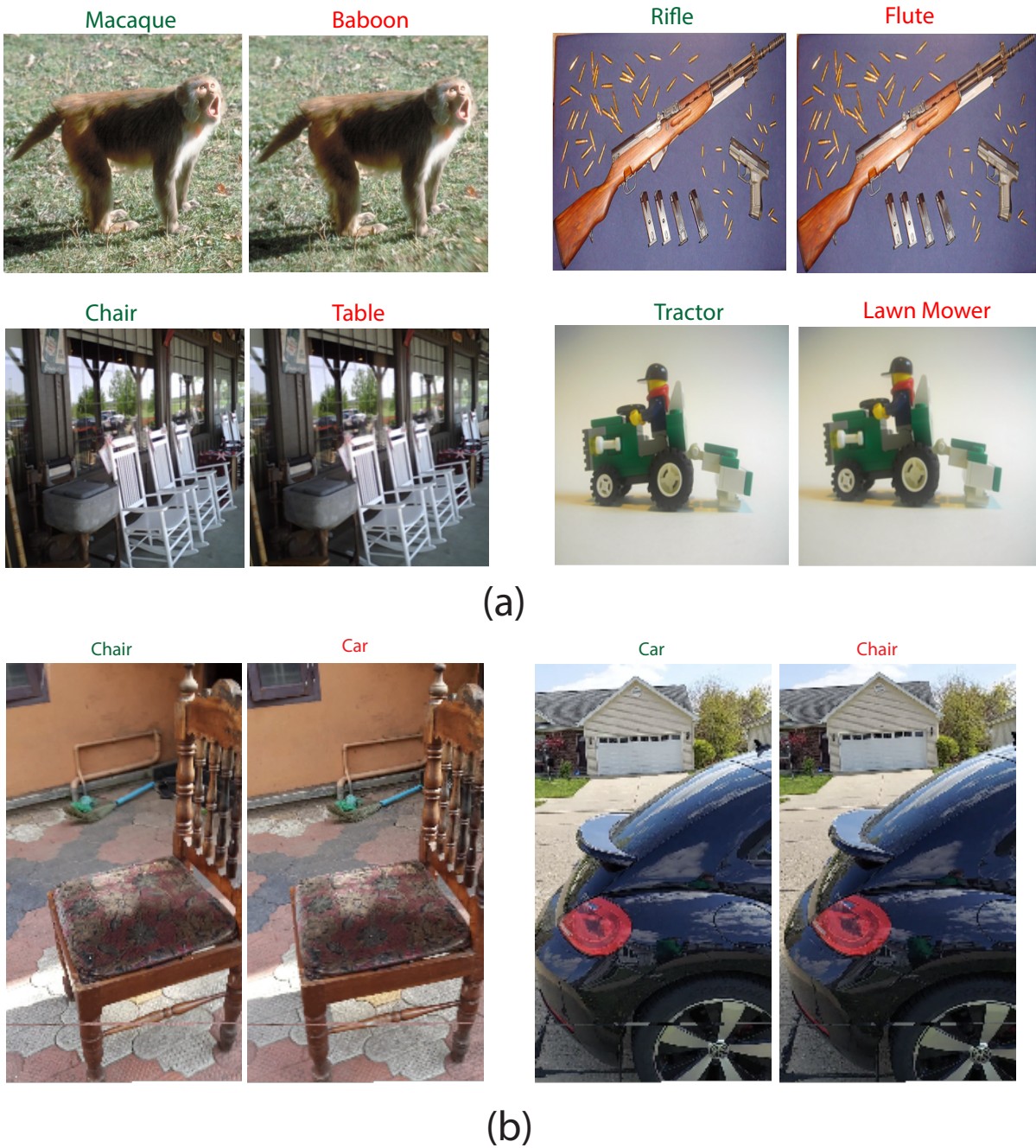

Figure 4: *In-distribution adversarial attacks on natural images.* (a) Misclassifications in ImageNet caused by CMA-Search + novel view synthesis. Examples are presented for a ResNet model trained on ImageNet, and OpenAI's CLIP model. (b) Sample errors for the Co3D dataset searched within $1 - 5$ frames of a correctly classified image. Attack rates for Co3D are reported in Table 2.

perturbations Chen et al. (2020); Lee et al. (2018); Roth et al. (2019); Hendrycks & Gimpel (2016). Second, and more concerning, they offer perfect plausible deniability to malicious actors. Unlike traditional attacks that leave forensic evidence of pixel manipulation, in-distribution adversarial examples are indistinguishable from legitimate data. Consider a self-driving car crash due to misclassifying a natural scene—not only does it bypass security systems as a perfectly natural image, but it also becomes impossible to determine whether this scene was deliberately chosen to cause failure or was truly accidental. This fundamentally undermines security approaches that rely on detecting tampering or establishing malicious intent.

With CMA-Search, we have presented a tool to help future researchers search for such failures and evaluate such defense mechanisms. Naturally, a model with no in-distribution failures would have an attack rate of 0, making this a natural goal for benchmarking studies using this metric. Future research could also extend this to include distance-constrained variants, such as Attack Rate at 1% or 5%, which would measure vulnerability within specific perturbation bounds.

Several promising mitigation strategies emerge from our analysis, which could guide future work on in-distribution robustness. Firstly, our results in Fig. 2(c) show these errors cluster near category boundaries. This suggests that targeted sampling strategies which densely sample training points in these regions could improve model robustness. Secondly, we also found that model initialization has a profound impact on the attack rate, which warrants further research into identifying better initialization strategies which can lead to more robust models. Finally, drawing inspiration from biology, the saccadic movements of the human eye suggest that using multiple viewpoints of the same scene during test time could reduce vulnerability.

## 9 Acknowledgements

We are grateful to George Alvarez and Talia Konkle for their encouragement and for highlighting the relevance of our findings to the fields of Psychology and Vision Sciences. We also extend our thanks to Utkarsh Singhal, Pranav Misra, Fenil Doshi, and Elisa Pavarino for their insightful discussions and feedback. This research was partially supported by Fujitsu Limited (Contract No. 40009105), NSF grants CRCNS-2309041, IIS-2127544, and NIH grant R01HD104969.

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

## S1 Graphics pipeline to generate dataset of camera and lighting variations

### S1.1 3D Scene Setup

Each scene contains one camera, one 3D model and 1-4 lights. To ensure no spuri- ous correlations with object texture [17], texture for all ShapeNet objects was replaced with a simple diffuse material and the background was kept constant to ensure no spurious correlations between foreground and background. Thus, every scene is completely parametrized by the camera and the light parameters. As shown in Fig. 1, camera parameters are 10 Dimensional: one dimension for the FOV (field of view of camera lens), and three dimensions each for the Camera Position (coordi- nates of camera center), Look At (point on the canvas where the camera looks), and the UP vector (rotation of camera). Analogously, lights are represented by 11 dimensions - two dimensions for the Light Size, and three each for Light Position, light Look At and RGB color intensity. Multiple lights ensure that scenes contain complex mixed lighting, including self-shadows. Thus, our scenes are $(11n + 10)$ dimensional, where n is the number of lights. There is a one-to-one mapping between the pixel space (rendered images) and this low dimensional scene representation.

### S1.2 Unbiased, uniform sampling of scene parameters

To ensure an unbiased distribution over different viewpoints, locations on the frame, perspective projections and colors, we ensured that scene parameters follow a uniform distribution. Concretely, camera and light positions were sampled from a uniform distribution on a spherical shell with a fixed minimum and maximum radius. The Up Vector was uniformly distributed across range of all possible camera rotations, and RGB light intensities were uniformly distributed across all possible colors. Camera and light Look At positions were uniformly distributed while ensuring the object stays in frame and is well-lit (frame size depends on Camera Position and FOV). Finally, Light Size and camera FOV were uniformly sampled 2D and 1D vectors. Hyper-parameters for rendering, along with the exact distribution for each scene parameter and the corresponding sampling technique used to sample from these distributions are reported in the supplement.

Below we specify the hyper-parameters for rendering, along with the exact distribution for each scene parameter and the corresponding sampling technique used to sample from these distributions.

**Camera Position:** For scene camera, first a random radius $r_c$ is sampled while ensuring $r_c \sim \text{Unif}(0.5, 8)$. Then, the camera is placed on a random point denoted $(x_c, y_c, z_c)$ on the spherical shell of radius $r_c$. To generate a random point on the sphere while ensuring an equal probability of all points, we rely on the method which sums three randomly sampled normal distributions Harman & Lacko (2010):

$$X, Y, Z \sim \mathcal{N}(0, 1), \tag{3}$$
$$v = (X, Y, Z), \tag{4}$$
$$(x_c, y_c, z_c) = r_c * \frac{v}{\|v\|}. \tag{5}$$

**Camera Look At:** To ensure the object is shown at different locations within the camera frame, the camera Look At needs to be varied. However, range of values such that the object is visible can be present across the entire range of the frame depends on the camera position. So, we sample camera Look At as $l_c$ as follows:

$$l_c \sim \text{Unif}(K * x_c, K * y_c, K * z_c), \text{where } K = 0.3. \tag{6}$$

The value $K = 0.3$ was found empirically. We found it helped ensure that objects show up across the whole frame while still being completely visible within the frame.

**Camera Up Vector:** Note that the camera Up Vector is implemented as the vector joining the camera center (0,0,0) to a specified position. We sample this position and therefore the Up Vector $u_c$ as follows:

$$x, y, z \sim \text{Unif}(-1, 1), \tag{7}$$
$$u_c = (x, y, z). \tag{8}$$

**Camera Field of View (FOV):** We sample the field of view $f_c$ while ensuring:

$$f_c \sim \text{Unif}(K_1, K_2). \tag{9}$$

Again, the values $K_1 = 35, K_2 = 100$ were found empirically to ensure objects are completely visible within the frame while not being too small.

**Light Position:** For every scene we first sample the number of lights $n$ between 1-4 with equal probability. For each light $i$, a random radius $r_i$ is sampled ensuring $r_i \sim \text{Unif}(R_1, R_2)$, then the light is placed on a random point $(x_i, y_i, z_i)$ on the sphere of radius $r_i$. $R_1 = 1$ and $R_2 = 8$ were found empirically to ensure that the light is able to illuminate the 3D model appropriately.

**Light Look At:** To ensure that the light is visible on the canvas, light Look At is sampled as a function of the camera position:

$$l_i \sim \text{Unif}(K * x_c, K * y_c, K * z_c), \text{where } K = 0.3. \tag{10}$$

As in the case of the Camera Look At parameter mentioned above, the value $K = 0.3$ was found empirically.

**Light Size:** Every light in our setup is implemented as an area light, and therefore requires a height and width to specify the size. We generate the size $s_i$ for light $i$ as:

$$h, w \sim \text{Unif}(L_1, L_2), \tag{11}$$
$$s_i = (h, w). \tag{12}$$

$L_1 = 0.1, L_2 = 5$ were found empirically to ensure the light illuminates the objects appropriately.

**Light Intensity:** This parameter specifies the RGB intensity of the light. For light $i$, RGB color intensity $c_i$ was sampled as:

$$r, g, b \sim \text{Unif}(0, 1), \tag{13}$$
$$c_i = (r, g, b). \tag{14}$$

**Object Material:** To ensure no spurious correlations between object texture and category, all object textures were set to a single diffuse material. Specifically, the material is a linear blend between a Lambertian model and a microfacet model with Phong distribution, with Schlick's Fresnel approximation. Diffuse reflectance was set to 1.0, and the material was set to reflect on both sides.

### S1.3 3D models used for generating two different test sets

Our dataset contains 11 categories, with 40 3D models for every category chosen from ShapeNet Chang et al. (2015). Neural networks were evaluated on two test sets - one with the 3D models seen during training, and the second with new, unseen 3D models. The first test set was generated by simply repeating the same procedure as described above. Thus, the *(Geometry × Camera × Lighting)* joint distribution matches exactly for the train set and this test set. The second test set was created by the exact same generation procedure, but with 10 new 3D models for every category chosen from ShapeNet. The motivation for this second test set was to ensure our models are not over-fitting to the 3D models used for training. Thus, the *(Camera × Lighting)* joint distribution matches exactly for this test set and the train set, but the *Geometry* is different in these two sets.

## S2 Generating nearby views for Natural Image Datasets

### S2.1 Views in the vicinity of ImageNet images

ImageNet contains only one viewpoint per object. While several variations of ImageNet have been proposed by adding noise in the form of corruptions and perturbations Hendrycks & Dietterich (2019), these variations are

Table S3: Performance of object recognition models on seen and new 3D models.

| Accuracy | ResNet | Anti-Aliased | Truly Shift Invariant | ViT | DeIT | DeIT Distilled |
|---|---|---|---|---|---|---|
| Seen models | 0.75 | 0.82 | 0.80 | 0.58 | 0.63 | 0.64 |
| New models | 0.70 | 0.74 | 0.72 | 0.59 | 0.64 | 0.65 |

designed to study the impact of out-of-distribution shifts on object recognition models. Like these variations, our camera manipulations correspond to transforming input images to study its impact on object recognition models. However, the key difference is that our work focuses on in-distribution adversarial examples, due to which these datasets designed for out-of-distribution shifts cannot be repurposed for our experiments. Thus, a major challenge in extending our results to ImageNet is generating natural images in the vicinity of a correctly classified image by slightly modifying the camera parameters. To do so for ImageNet is equivalent to novel view synthesis (NVS) from single images, which has been a long-standing challenging task in computer vision. However, recent advances in NVS enable us to extend our method to natural image datasets like ImageNet Yoon et al. (2020); Zhou et al. (2016); Wiles et al. (2020); Tucker & Snavely (2020).

To generate new views in the vicinity of ImageNet images, we rely on a single-view synthesis model based on multi-plane images (MPI) Tucker & Snavely (2020). The MPI model takes as input an image and the $(x, y, z)$ offsets which describe camera movement along the X, Y and Z axes. Note that unlike our renderer, it cannot introduce changes to the camera Look At, Up Vector, Field of View or lighting changes. An important limitation of this approach is that any noise added by the MPI model in image generation is a confounding variable which we cannot account for. This further highlights the importance of our rendered and Co3D experiments as these experiments do not suffer from such noise.

### S2.2 Views in the vicinity of Co3D images

As an additional control for any potential noise introduced by the novel view synthesis pipeline in generating nearby views for ImageNet images, we present additional results on the large-scale, multi-viewpoint Co3D Reizenstein et al. (2021) dataset. Co3D was created by capturing short videos of fixed objects placed on a surface by a user moving a mobile phone around the object. Thus, nearby frames in the video represent views in the vicinity of an image. We utilize this to test in-distribution robustness in the vicinity of correctly classified images. The classification dataset is created by picking 5 categories—car, chair, handbag, laptop, and teddy bear. We created the training data by uniformly sampling frames across the whole video for all videos for these categories amounting to $187, 200$ training images. Note that this amounts to roughly $38, 000$ images per category, which is 32 times the ImageNet training set on a per category basis. An in-distribution test set of $68, 854$ images is generated by sampling the remaining frames to measure overall accuracy of the trained models. We then search for in-distribution failures in the vicinity (i.e., nearby frames) from the remaining frames from these videos in the Co3D dataset. Thus, no novel view synthesis pipeline was used. Instead, pre-captured frames from the videos were used to search for in-distribution adversarial examples in the vicinity of viewpoints.

## S3 Additional details on CMA-Search

Below we provide details on the implementation and evaluation of our in-distribution adversarial search method—CMA-Search.

### S3.1 Finding in-distribution adversarial examples by searching the vicinity of a correctly classified image

CMA-Search can be used to attack any parametric dataset. To find an in-distribution failure our methodology requires a classification model, a correctly classified data point, and the parametric representation of this data point. CMA-Search optimizes these parameters using evolutionary strategies to find a sample which is misclassified by the model. We used a gradient-free optimization method—Covariance Matrix Adaptation-Evolution Strategy (CMA-ES) Hansen & Ostermeier (1996); Hansen (2016). CMA-ES has been found to

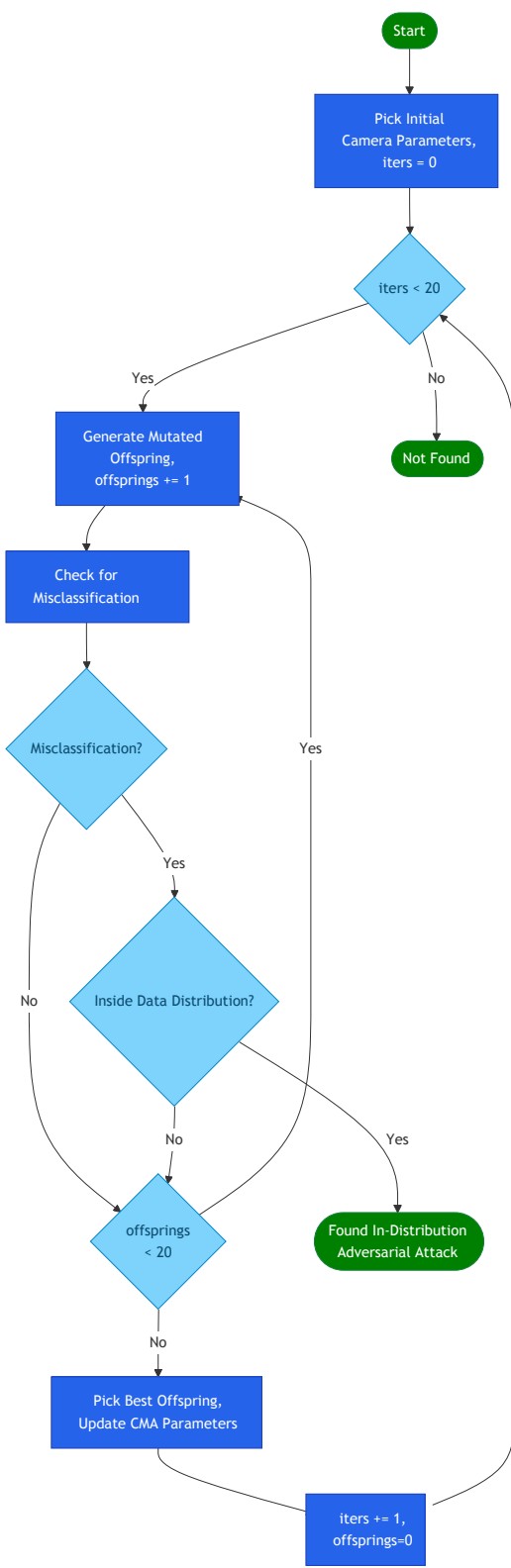

Figure S1: *Flowchart describing CMA-Search step-by-step.*

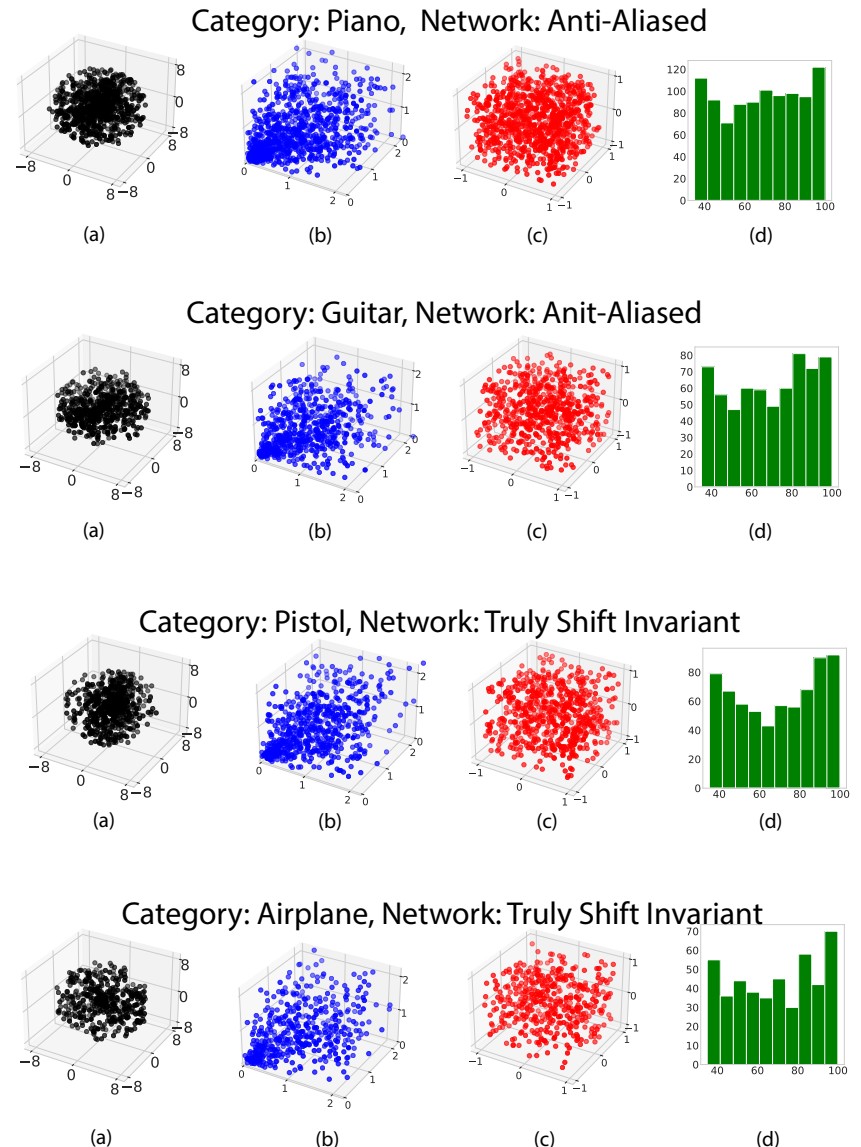

Figure S2: *Camera Parameters that lead to misclassifications for multiple categories and architectures.* (a) Camera Position, (b) Camera Look At, (c) Up Vector, (d) Histogram of Lens Field of View.

work reliably well with non-smooth optimization problems and especially with local optimization Hansen & Ostermeier (2001), which made it a perfect fit for our search strategy.

Starting from the initial parameters, CMA-ES generates offspring by sampling from a multivariate normal (MVN) distribution i.e. mutating the original parameters. These offspring are then sorted based on the fitness function (classification probability), and the best ones are used to modify the mean and covariance matrix of the MVN for the next generation. The mean represents the current best estimate of the solution i.e. the maximum likelihood solution, while the covariance matrix dictates the direction in which the population should be directed in the next generation. The search was stopped either when a misclassification occurred, or after 15 iterations over scene parameters. For the simplistic parametrically controlled data, we checked for a misclassification till 1500 iterations.

For ease, we present the algorithm for in-distribution errors in rendered images found by optimizing camera parameters. The methodology to attack all datasets is analogous. In this case, our method searches the vicinity of the camera parameters of a correct classified image to find an in-distribution error. Algorithm 1 provides an outline of using CMA-Search to find in-distribution adversarial examples by searching the vicinity of camera parameters. The algorithm for searching for adversarial examples using light parameters in rendered data, and within parametrically controlled uniform data is analogous. Fig. 3(c) presents examples of in-distribution adversarial examples found using CMA-Search over camera parameters. As shown, subtle changes in 3D perspective can lead to drastic errors in classification. We also report the subtle changes in camera position (in black) and camera Look At (in blue) between the correctly and incorrectly classified images in Fig. 3(c).

This approach differs from existing work on adversarial viewpoints and lightingLiu et al. (2019); Zeng et al. (2019); Jain et al. (2019) in two ways. First, unlike these works our approach finds in-distribution errors. Secondly, these methods rely on gradient descent and thus require high dimensional representations of the scene to work well. For instance, these works often use neural rendering where network activations act as a high dimensional representation of the scene Zeng et al. (2019); Joshi et al. (2019), or use up-sampling of meshes to increase dimensionality Liu et al. (2019). In contrast, our approach works well for as low as 3 dimensions.

### S3.2  Evaluating CMA-Search and in-distribution robustness using the Attack Rate

The performance of CMA-Search was quantified using a new metric—the *Attack Rate*, which refers to the percentage of correctly classified points for which CMA-Search successfully found an in-distribution adversarial example. For simplistic parametrically controlled data, the *Attack Rate* was measured by attacking $20,000$ correctly classified samples using CMA-Search. Due to our use of a physically based renderer that accurately models the physics of light in the 3D scene, generating images in the vicinity of the correctly classified image is a computational intensive process. Thus, for rendered data, the *Attack Rate* is measured by attacking $2,000$ correctly classified images for every architecture, and these numbers are reported in Table 1. As an additional control, we also measured the *Attack Rate* for the ResNet18 architecture with $20,000$ images, and found the rate to be unchanged. For the Co3D dataset, *Attack Rate* is measured on $116,850$ images. As explained in Sec.3, we do not render or generate any new novel views for Co3D but simply search through natural images already provided in the dataset.

### S3.3  Visualizing in-distribution adversarial examples using Church-window plots

CMA-Search starts from a correctly classified point and provides an in-distribution adversarial example. We used these two points to define a unit vector in the adversarial direction, and fixed this as one of basis vectors for the space the data occupies. As data dimensionality was $D$, we calculated the remaining $D-1$ orthonormal bases. Following the same protocol as past work Warde-Farley & Goodfellow (2016), we randomly picked one of these orthonormal vectors as the orthogonal direction and defined a grid of perturbations with fixed increments along the adversarial and the orthogonal directions. These perturbations were then added to the original sample and the model was evaluated at these perturbed samples. We plotted correct classifications in white, in-distribution adversarial examples in red, and out-of-distribution samples in black.

### S3.4  Computational efficiency of CMA-Search

CMA-Search operates iteratively, generating multiple offsprings in every iteration, and retaining the best in every iteration to calculate parameters for the next iteration. For simplistic parametrically controlled, CMA-Search was set to generate 20 offsprings in every iteration, and the search algorithm was set to stop when an in-distribution adversarial example is found, or if a maximum threshold of 1500 iterations were hit. On average, 51 iterations were needed to find an in-distribution adversarial example for 10 dimensional data. The average number of iterations needed dropped to 20 for 100 dimensional data. Note that as dimensionality increases, all steps become more computationally intensive, this includes training models, generating new offsprings using CMA-Search, and model inference to test offspring fitness. Thus, overall time required

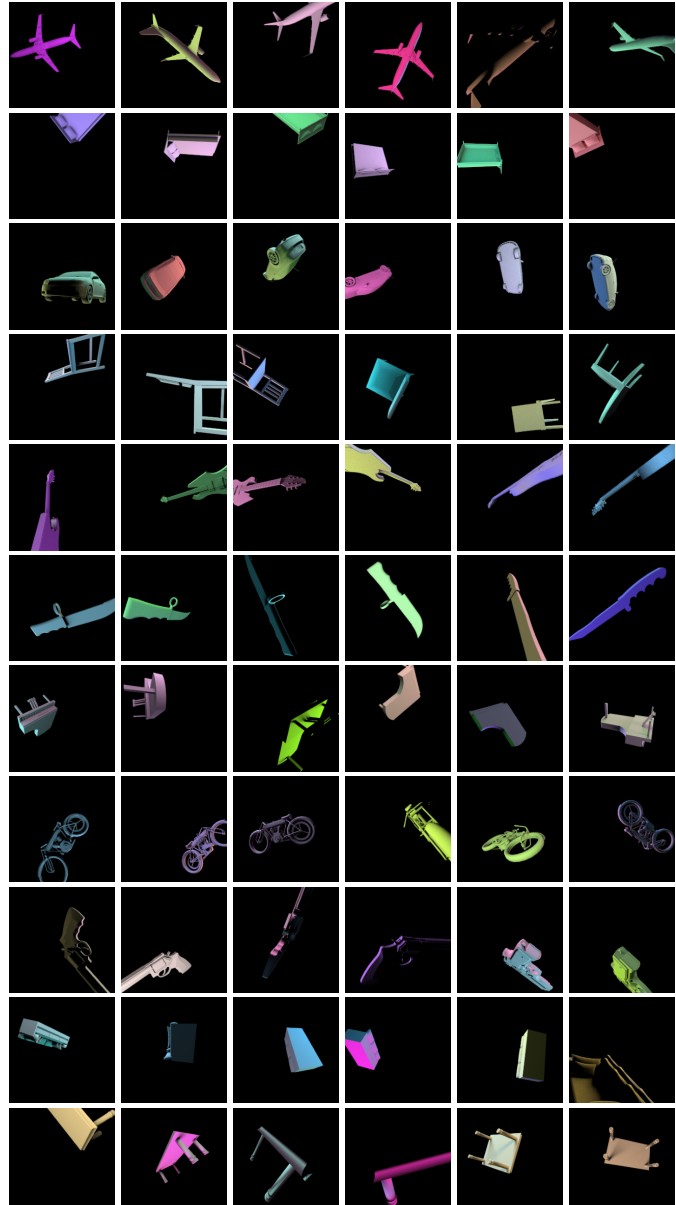

Figure S3: *Sample Images from our rendered dataset.*

to attack increases with dimensionality. However, computational efficiency of CMA-Search improves with dimensionality, as lesser iterations are needed.

For rendered data, which is significantly higher dimensional, we found that CMA-Search is very efficient as extremely low number of iterations are needed to find an in-distribution failure. For both camera and light variation based attacks, CMA-Search was set to generate 10 offsprings in every iteration, and maximum iteration threshold was set to 15. On average, only 2 iterations were needed to find an in-distribution failure with camera variations. For light variations, 3.5 iterations were required on average. This suggests that CMA-Search is more efficient at higher dimensions, despite working well at low dimensions.

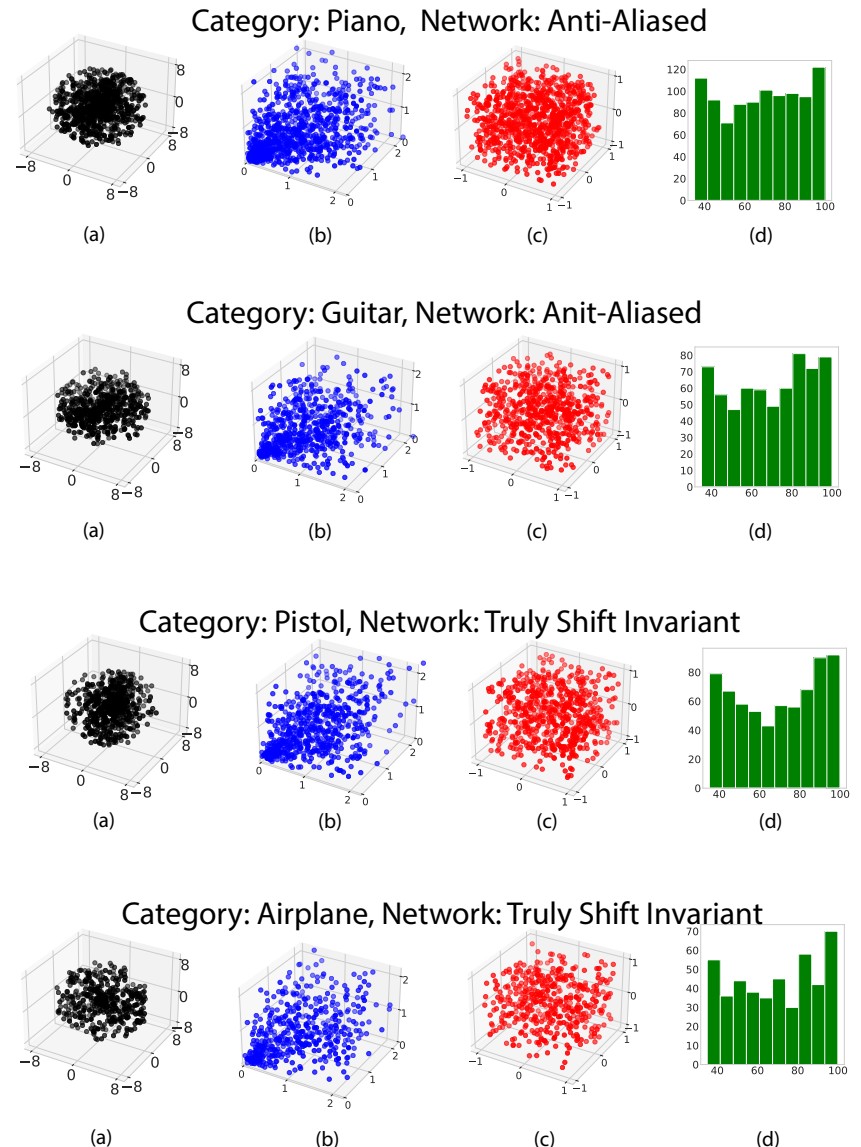

Figure S4: *Camera Parameters that lead to misclassifications for multiple categories and architectures.* (a) Camera Position, (b) Camera Look At, (c) Up Vector, (d) Histogram of Lens Field of View.

## S4  Additional Results with ImageNet

We present examples of in-distribution adversarial attacks with ImageNet in Fig. S5. Furthermore, we conducted comprehensive analysis on how noise levels impact robustness. Specifically, we evaluated the attack rate for ImageNet images under different levels of noise introduced by NVS. A total of 27 noise levels were tested—three levels each for the camera parameters controlling the X, Y, and Z axes (corresponding to horizontal translation, vertical translation, and zoom). These results are presented in Fig. S6. While attack rates increased with noise across all axes, models showed particular susceptibility to Y-axis perturbations.

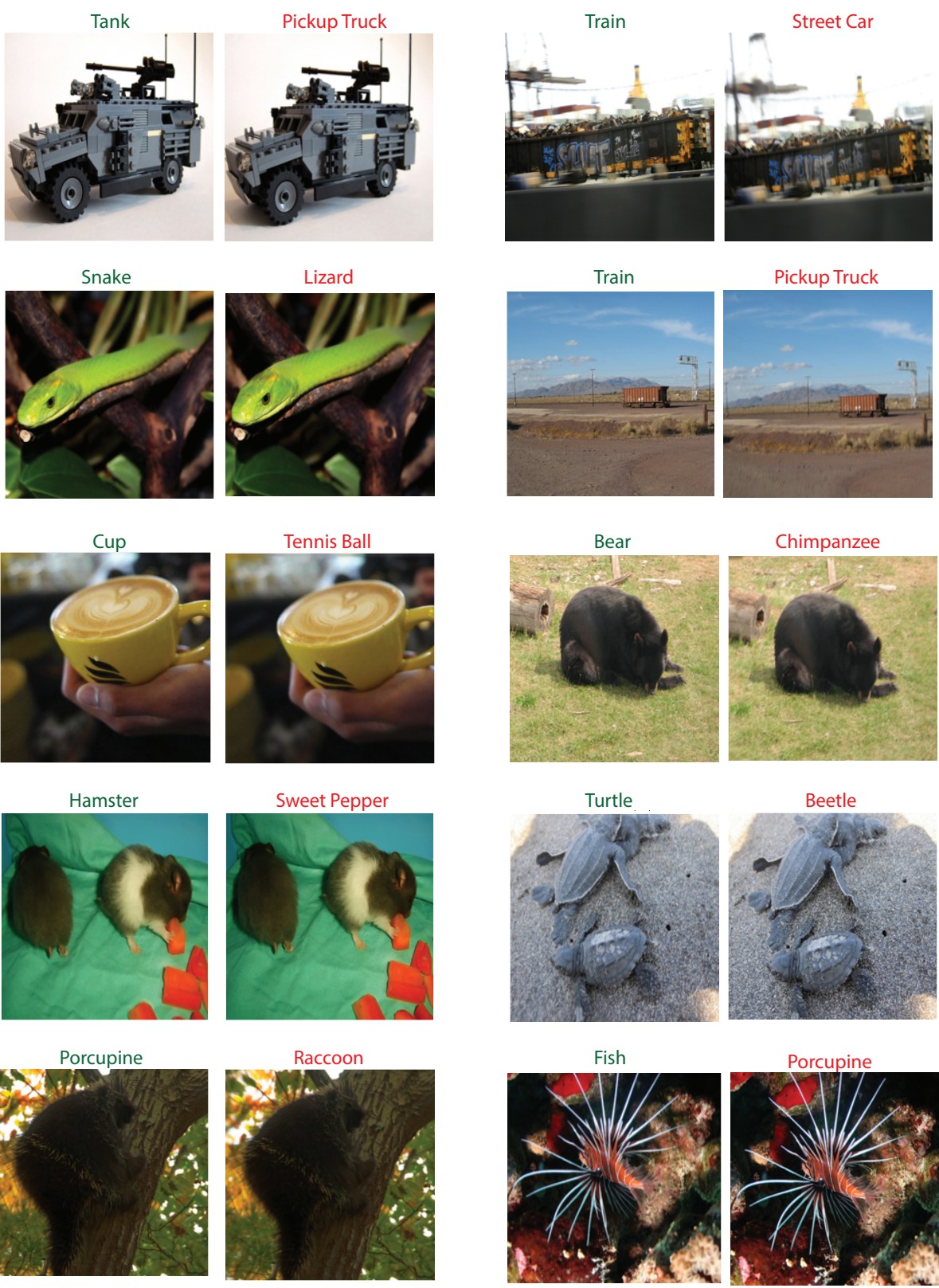

Figure S5: *ore examples of misclassified ImageNet-like images discovered by CMA-Search combined with the single view MPI model.*

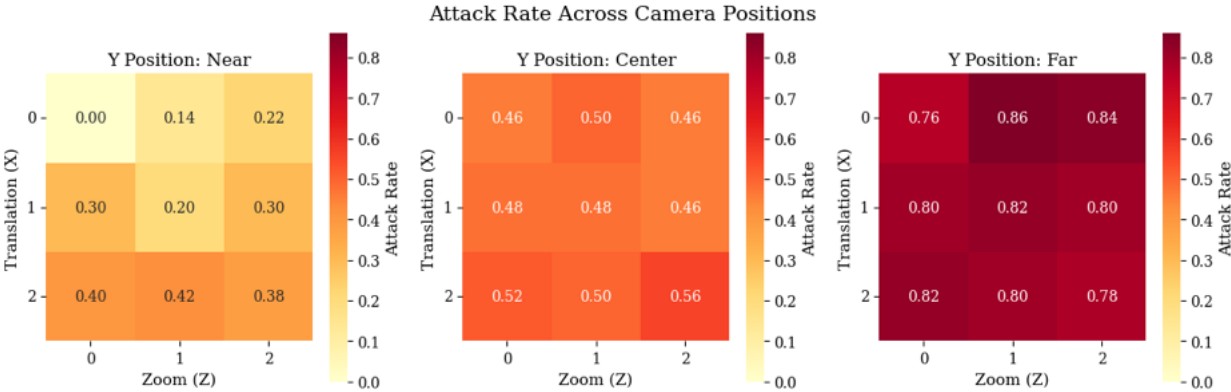

Figure S6: *Impact of NVS induced noise on ImageNet classification.*

