# OpenReview forum: "In-distribution adversarial attacks on object recognition models using gradient-free search."
_TMLR — Accepted by TMLR_

### Review · Reviewer_mN3d · 2024-10-16

**Summary Of Contributions:**

The main contribution of this paper is to provide the evidence for the existence of in-distribution adversarial examples using real-world data. In-distribution adversarial example is the sample within the training data distribution, but causes severe degrade in model's performance.
The empirical evidence is provided through a series of experiments using various datasets from artificially generated toy data to real datasets including Co3D.
The experimental results show that we can construct a set of inputs that cause a significant decrease in accuracy from the training data distribution even for models with high accuracy on test data.

**Audience:**

Yes

**Claims And Evidence:**

Yes

**Requested Changes:**

Please see weaknesses above.

**Strengths And Weaknesses:**

## Strength
- They provide empirical evidence of the existence of in-distribution adversarial examples against deep neural networks trained on real-world data.
- The main claim of this paper is adequately supported through experiments using multiple datasets, including real-world datasets.

## Weakness
- While this paper provides empirical evidence for the existence of in-distribution adversarial examples, it does not involve theoretical analysis. Therefore, it is unclear whether the existence of in-distribution adversarial samples is an unavoidable problem.
- For the same reason, it is not clear whether there are any in-distribution adversarial samples for data from domains other than images.
- Is there a mathematical definition of "in-distribution"? In the case of Co3D, I can understand the concept of in-distribution from the perspective of how the adversarial example is taken. However, I am not sure that the results of Novel View Synthesis for ImageNet are within the distribution of the training data.
### minor
- Reference on p.7, in the paragraph ``Visualizing failures'', is broken.

---

> ### Author Response · Authors · 2024-11-01
> **Thank you for your vote of confidence and valuable feedback.**
>
> Dear Reviewer mN3d,
>
> Thank you so much for your valuable feedback, and for highlighting that our experiments adequately support the main claim of the paper. Below we respond to your comments:
>
> 1. **No theoretical analysis:** As highlighted in Sec.2 of the paper, several theoretical works have already shown the presence of in-distribution adversarial attacks in theory [1,2,3,4]. Our main contribution here is showing that this phenomenon extends beyond theory—we confirm in-distribution adversarial attacks on object recognition models. This was confirmed using the image data of real-world objects, and the natural image datasets in our work. While theoretical works have shown the presence with simple, parametric data, we extend these results to natural images and object recognition.
>
> 2. **It is unsure if In-distribution attacks extend to other domains:**  Sec.3.1 confirmed that this phenomenon is not specific to image data. The experiments in this section showed in-distribution adversarial attacks on data which were not images.
>
> 3. **Mathematical definition of in-distribution:** Thank you for raising this point. Mathematically, a sample $x^*$ is defined to be in-distribution w.r.t. a dataset $X=\\{x_1,...x_N\\}$, if $x^*$ and all points $(x_1...x_N)$ are generated by sampling i.i.d. from the same generative distribution. Thus, as $n \to \infty$, $x^* \in X$. As an example, consider the Camera Position. Our dataset with camera and lighting variations was constructed by rendering images with camera position uniformly sampled from $[0.5, 8]$ units. Thus, any image of a scene with camera position within the range $[0.5,8]$ is in-distribution. Images with camera position not in this range are considered out-of-distribution. With this principle, we present controlled datasets at three levels of complexity which are used in our experiments. We have now incorporated this definition into the manuscript based on your suggestion.
>
> 4. **ImageNet results:** We agree that confirming whether Novel View Synthesis (NVS)-generated images are truly in-distribution for ImageNet is challenging. While we can accurately define the data distribution for datasets like Co3D, we can only make approximations for ImageNet’s data distribution. However, we have included them due to ImageNet’s widespread use in the AI research community. We believe that validating findings on this dataset has practical importance.
>
> 5. **Broken Reference:** Thank you for pointing this out; we have corrected it in the revised manuscript.
>
> ***References***
>
> 1. Justin Gilmer, Luke Metz, Fartash Faghri, Samuel S Schoenholz, Maithra Raghu, Martin Wattenberg, and Ian Goodfellow. The relationship between high-dimensional geometry and adversarial examples. arXiv preprint, arXiv:1801.02774, 2018.
>
> 2. Alhussein Fawzi, Seyed-Mohsen Moosavi-Dezfooli, and Pascal Frossard. Robustness of classifiers: from adversarial to random noise. In Advances in Neural Information Processing Systems, volume 29, 2016.
>
> 3. Alhussein Fawzi, Hamza Fawzi, and Omar Fawzi. Adversarial vulnerability for any classifier. In Advances in Neural Information Processing Systems, volume 31, 2018.
>
> 4. Alhussein Fawzi, Omar Fawzi, and Pascal Frossard. Analysis of classifiers’ robustness to adversarial perturbations. Machine learning, 107(3):481–508, 2018.

---

> > ### Author Response · Authors · 2024-12-02
> > **Following up on Rebuttal Submission.**
> >
> > Dear Reviewer,
> >
> > Thank you once again for your valuable feedback. We believe the revisions comprehensively address the reviewers' concerns and significantly strengthen the evidence supporting our primary claim: "Classification models, including object recognition models, are deeply susceptible to in-distribution adversarial attacks, even when achieving near-perfect test set accuracies."
> >
> > Please let us know if there are any additional changes or clarifications that would further improve the manuscript.
> >
> > Thank you for your time and consideration.

---

### Review · Reviewer_ioAP · 2024-10-21

**Summary Of Contributions:**

The paper studies the existence and generation of "in-distribution adversarial examples" for object recognition models. For this, the authors use a gradient-free adversarial attack that is based on the CMA-ES algorithm. The authors find interesting in-distribution adversarial examples when varying camera and light on 3d models from ShapeNet as well when generating novel views for ImageNet and Co3D datasets.

**Audience:**

Yes

**Broader Impact Concerns:**

no concerns

**Claims And Evidence:**

No

**Requested Changes:**

Please address the points listed under weaknesses. In particular, a term so central to the paper like "in-distribution" requires a clear definition.

**Strengths And Weaknesses:**

Strengths
 - Understanding failure cases of the model that are "in-distribution" is a problem of great practical importance as it is not only relevant in the presence of an adversary but also more broadly for any deployment scenario
 - The studied domains of controlling camera and light variations on ShapeNet (Section 3.2) and Novel View Synthesis on ImageNet are sufficiently closed to realistic problems to be of practical interest.
 - Figure 1 is a good illustration of the problem setting and makes the paper more accessible
 - Figure 3 provides relevant findings of in-distribution adversarial examples
 - the paper is overall well-written and easy to follow

Weaknesses
 - ID vs OOD: in general, the notions of in-distribution vs. out-of-distribution are central for the paper but never clearly defined. Often, "in-distribution" is used in a way as if the region (support) of the distribution is meant, for instance when using formulations like "search _inside_ this data distribution" or drawing a region in Figure 1. However, even a simple 1D Normal distribution has the whole real numbers as support - so everything would be "inside the data distribution" for a Normal distribution. The authors need to be more explicit about what they understand as "in-distribution" without making the concept too broad and at the same time show that their method finds actual in-distribution adversarial examples under this definition
 - CMA-Search is a straightforward application of CMA-ES to adversarial example generation with practical limitations - for instance, how would this method deal with constraints on the parameter ranges?
 - It remains unclear to me how constraints of the problem/threat model are enforced - for instance, where do the authors enforce that adversarial examples on the Nd uniform distribution data from Section 3.1 actually stay within the prescribed intervals?
 - Please clarify how in-distribution and out-of-distribution are determined in Figure 2c - from  Section 3.1 my understanding would be that everything with a value between a=-10 and b=10 is in-distribution and belongs to class y_i=0
 - Related work is very short and shallow.

---

> ### Author Response · Authors · 2024-11-01
> **Clearly defining In-Distribution attacks and clarifying questions raised in the review.**
>
> Dear Reviewer ioAP,
>
> Thank you for your valuable feedback and for recognizing the practical significance of our work. We appreciate your insights, particularly regarding the need for a clear definition of in-distribution adversarial attacks. We hope that the clear definition of In-Distribution adversarial attacks provided below clarifies the main claim of the paper—*object recognition models are plagued by in-distribution adversarial failures*, and showcases how our experiments across several different datasets provide corroborative evidence for this claim.
>
> **1. Clearly defining InD vs OOD:**
>
> You are absolutely correct.  We used the support of the distribution to define in and out-of-distribution samples. While it would be possible to construct two distributions with the same support and different PDFs, throughout this paper we focus on a simplified case—we always sample uniformly across the support in all our datasets. This choice allows the support to uniquely characterize the data distribution.
>
> &nbsp;&nbsp;&nbsp;&nbsp; **We have added the text below to Sec.3, which provides a precise mathematical definition for InD vs OOD:**
>
> &nbsp;&nbsp;&nbsp;&nbsp; Mathematically, a sample $x^*$ is defined to be in-distribution w.r.t. a dataset $X=\\{x_1,...x_N\\}$, if $x^*$ and all points $(x_1...x_N)$ are generated by sampling i.i.d. from the same generative distribution. Thus, as $n \to \infty$, $x^* \in X$. As an example, consider the Camera Position. Our dataset with camera and lighting variations was constructed by sampling rendering images with camera position uniformly sampled from $[0.5, 8]$ units. Thus, any image of a scene with camera position within the range $[0.5,8]$ is in-distribution. Images with camera position not in this range are considered out of distribution. For all datasets, we sample uniformly across the support.  This choice allows the support to uniquely characterize the data distribution.
>
> &nbsp;&nbsp;&nbsp;&nbsp; **We have also added the following to Sec.4 where we describe CMA-Search:**
>
> &nbsp;&nbsp;&nbsp;&nbsp; CMA-ES is an unconstrained optimization procedure. Thus, we ensured that a misclassification counts as an in-distribution adversarial attack only if it met both criteria—(1) it caused a misclassification, (2) it belongs to the training data distribution (i.e. inside the support of the underlying distribution).
>
> **2. “CMA-ES is simple—how to handle constraints on parameter ranges”:**
>
> Our main contribution is showing that in-distribution adversarial attacks on object recognition models are possible. We were able to achieve this using unconstrained optimization using CMA-ES as showcased in our results. For this purpose, we believe that the simplicity of our optimization is a feature, and not a bug. While we achieved significantly high attack rates with CMA-ES, we hope that future work can incorporate better optimization strategies (including constrained optimization) to further improve the attack rates.
>
> **3. “How are the constraints enforced?”:**
>
> We have updated Algorithm 1 in the paper to clarify this. Simply put, whenever CMA-Search finds a misclassification, we check if the point belongs to the support of the training data distribution. If yes, the image is an in-distribution failure. Otherwise, we continue the iterative search. If no failure is found after a fixed number of iterations, we consider the data point safe from CMA-Search. Despite the simplicity, this approach results in very high attack rates.
>
> **4. “Clarifying how InD and OOD are defined in Fig 2c”:**
>
> You are absolutely correct. We clarify this further below, denoting the data point with $x$, and the predicted label with $y’$.
> - White points (InDist, correctly classified): $x_i \in [-10,10], y'_i=0$
> - Red points (In Dist failures): $x_i \in [-10,10], y'_i=1$
> - Black points (OOD): $x_i \not \in [-10,10]$. We did not report $y'_i$ for these points as they are irrelevant to our study.
>
> **5. Related Work:**
>
> Could you please advise what additional topics you would like us to cover? Any specific works we have missed? We would be happy to connect our work to additional existing papers.
>
> Thank you for your time, and please let us know if you have additional questions.

---

> > ### Author Response · Authors · 2024-12-02
> > **Following up on Rebuttal Submission.**
> >
> > Dear Reviewer,
> >
> > Thank you once again for your valuable feedback. We believe the revisions and clarifications mentioned in our revision comprehensively address your questions, and significantly strengthen the evidence supporting our primary claim: *"Classification models, including object recognition models, are deeply susceptible to in-distribution adversarial attacks, even when achieving near-perfect test set accuracies."*
> >
> > Please let us know if there are any additional changes or clarifications that would further improve the manuscript.
> >
> > Thank you for your time and consideration.

---

> > > ### Comment · Reviewer_ioAP · 2024-12-02
> > > **Feedback**
> > >
> > > Dear authors,
> > > thanks for your feedback. I have taken it into account when submitting my final recommendation two weeks ago but forgot to post a reply to your response directly. Thanks again for clarifying my questions and taking my recommendations into account!

---

### Review · Reviewer_GKQk · 2024-11-05

**Summary Of Contributions:**

This paper presents CMA-Search, a gradient-free search method for finding in-distribution adversarial images. It is motivated by the claim that adversarial examples are seen and studied mostly as out-of-distribution inputs, but in-distribution inputs can be adversarial as well.

It quickly reaches its methods section, describing various datasets to test on. These include synthetic data generated to test adversarial robustness in a controlled manner. Next is rendered data from sim, giving more control over image parameters. And finally, natural images. It then presents the adversarial input-finding method itself: it uses covariance matrix adaptation-evolution strategy (CMA-ES) to find adversarial examples through perturbed camera and lighting parameters. It considers these within training distribution given that the adjusted parameter values are naturalistic. It samples and evaluates iteratively to find incorrectly labeled examples.

The paper then goes through its experiment details and results: various architectures including MLPs, CNNs, and transformers are tried. There is also standard variation of hyperparameters. The results show that in all datasets, adversarial images are found a lot of the time; in the simplest and most controlled dataset, they are found in every training setting.

The paper concludes by discussing the importance of understanding adversarial examples and what causes them.

**Audience:**

Yes

**Broader Impact Concerns:**

Adversarial examples are a security risk. This is not addressed int he paper. I don't think it needs extensive discussion, but some discussion on the implications of this project for threatening adversarial examples would be helpful.

**Claims And Evidence:**

Yes

**Requested Changes:**

- Discuss not only the mechanism of determining adversarial images but also how to deal with them. This could be a future work, but discussing it is useful even in this paper because the solutions may tie directly to the identificatoin method itself.
- For natural image experiments involving NVS, analyze how potential noise impacts results.

**Strengths And Weaknesses:**

## Strengths:
- This is an original and intelligent problem + method. The gradient-free search idea is clever and provokes thought about the mechanisms underlying these adversarial examples. It is very intuitive for this use-case.
- Experimental setup is solid:
  - Datasets are varied and well-chosen. It's good to have a mix of synthetic datasets that highlight the mechanism in question, and large natural datasets that show that it's a real mechanism. Datasets are highly varied and diverse.
  - Good use of transformer-based architectures along with ResNets. Architecture has the potential to change the adversarial example landscape considerably, so it's helpful--even if intuitively expected--to see effects appear in different architectures. It also keeps this area of vision caught up with the state of the field.
- Good figures! The different visualization approaches help a lot to build our understanding of the underlying nature of these adversarial inputs, which is important not only for security but interpretability.

## Weaknesses:
- The paper motivates itself by talking about how adversarial failures mainly occur out of distribution. I'm not sure this is entirely true, though I do think it's fair to say most work in adversarial examples focuses on synthetic out-of-distribution examples.
- While the CMA-ES pseudocode is useful and important, it would also help to have a diagram of the method if possible. It's currently somewhat difficult to build intuition until rereading a few times.
- More grounding on metrics such as "attack rate" would be helpful - what should we look for and benchmark against?

---

> ### Author Response · Authors · 2024-11-22
> **Additional figures, results, and discussions have been added.**
>
> Dear Reviewer GKQk,
>
> Thank you for your valuable feedback, particularly your vote of confidence regarding our problem's originality and the rigour of our experimental evidence. We address your concerns and suggestions below:
>
> **1. "It would help to have a diagram of the method":**
>
> Thank you for the suggestion. We have added a new figure explaining the CMA-Search algorithm (See Fig.S1 added in the supplement, cited in Sec. 4). For your convenience here’s a link to the figure: https://drive.google.com/file/d/1S-FSLmmvQJz3mfyVqWww7jO-wU4RaSjj/view?usp=share_link.
>
> **2. "More grounding on metrics such as attack rate would be helpful":**
>
> As defined in Sec.4, attack rate represents the percentage of correctly classified points for which an in-distribution example could be found. A model with no in-distribution failures would have an attack rate of 0, making this a natural goal for benchmarking studies using this metric. Future research could extend this to include distance-constrained variants, such as Attack Rate at 1% or 5%, which would measure vulnerability within specific perturbation bounds. We have incorporated these discussions in the newly added Discussions section.
>
> **3. “For experiments involving NVS, analyse how potential noise impacts results”:**
>
> Following the suggestion, we have conducted comprehensive analysis on how noise levels impact robustness. Specifically, we evaluated the attack rate for ImageNet images under different levels of noise introduced by NVS. A total of 27 noise levels were tested—three levels each for the camera parameters controlling the X, Y, and Z axes (corresponding to horizontal translation, vertical translation, and zoom). These results have been added to the paper (See Fig.S6). For your convenience, here’s a link the figure: https://drive.google.com/file/d/10QhiYoXNXoAZv5LNhhT7UQX3ocmNxm8R/view?usp=share_link. While attack rates increased with noise across all axes, models showed particular susceptibility to Y-axis perturbations. These results are now included in the paper (See Fig.S6).
>
> **4. “Discuss not only the mechanism of determining adversarial images but also how to deal with them.”**
>
> Several promising mitigation strategies emerge from our analysis, which could guide future work on in-distribution robustness. Firstly, our results in Fig.2(c) show these errors cluster near category boundaries. This suggests that targeted sampling strategies which densely sample training points in these regions could improve model robustness. Secondly, we also found that model initialization has a profound impact on the attack rate, which warrants further research into identifying better initialization strategies which can lead to more robust models. Finally, drawing inspiration from biology, the saccadic movements of the human eye suggest that using multiple viewpoints of the same scene during test time could reduce vulnerability. We have added these to the newly added Discussion section.
>
> **5. "Some discussion on the implications of this project for threatening adversarial examples would be helpful."**:
>
> We agree with the reviewer and have added the below text to the newly added Discussions section:
> The discovery of in-distribution adversarial examples presents critical security challenges distinct from traditional adversarial attacks. While conventional attacks require engineered perturbations, these adversarial examples exist naturally within the expected data distribution, creating two severe vulnerabilities. First, they bypass traditional detection methods that rely on identifying out-of-distribution characteristics or unusual perturbations [1,2,3,4]. Second, and more concerning, they offer perfect plausible deniability to malicious actors. Unlike traditional attacks that leave forensic evidence of pixel manipulation, in-distribution adversarial examples are indistinguishable from legitimate data. Consider a self-driving car crash due to misclassifying a natural scene—not only does it bypass security systems as a perfectly natural image, but it also becomes impossible to determine whether this scene was deliberately chosen to cause failure or was truly accidental. This fundamentally undermines security approaches that rely on detecting tampering or establishing malicious intent.
>
> ***References***
>
> 1. Chen, J., Li, Y., Wu, X., Liang, Y. and Jha, S., 2020. Robust out-of-distribution detection via informative outlier mining. arXiv preprint arXiv:2006.15207, 1(2), p.7.
> 2. Roth, K., Kilcher, Y. and Hofmann, T., 2019, May. The odds are odd: A statistical test for detecting adversarial examples. In International Conference on Machine Learning (pp. 5498-5507). PMLR.
> 3. Lee, K., Lee, K., Lee, H. and Shin, J., 2018. A simple unified framework for detecting out-of-distribution samples and adversarial attacks. Advances in neural information processing systems, 31.
> 4. Hendrycks, D. and Gimpel, K., 2016. Early methods for detecting adversarial images. arXiv preprint arXiv:1608.00530.

---

> ### Author Response · Authors · 2024-12-02
> **Following up on Rebuttal Submission.**
>
> Dear Reviewer,
>
> Thank you once again for your valuable feedback. We believe the revisions and clarifications mentioned in our revision comprehensively address your questions, and significantly strengthen the evidence supporting our primary claim: *"Classification models, including object recognition models, are deeply susceptible to in-distribution adversarial attacks, even when achieving near-perfect test set accuracies."*
>
> Please let us know if there are any additional changes or clarifications that would further improve the manuscript.
>
> Thank you for your time and consideration.

---

### Author Response · Authors · 2024-11-29
**Thank you for your feedback! Please let us know if any additional questions.**

Dear Reviewers,

Thank you for your valuable feedback. For ease, we summarize below all changes made to the manuscript in response to your questions.

### 1. Mathematically defining In and Out-of-Distribution data  *[Addressing Reviewers ioAP and mN3d]*

Text added to Section 3: A sample $x^*$ is defined to be in-distribution with respect to a dataset $X=\{x_1,...x_N\}$ if $x^*$ and all points $(x_1...x_N)$ are generated by sampling i.i.d. from the same generative distribution. Thus, as $n \to \infty$, $x^* \in X$. To illustrate this with a concrete example from our work: for the Camera Position dataset, images are rendered with camera positions uniformly sampled from $[0.5, 8]$ units. Therefore, any image with camera position within $[0.5, 8]$ is in-distribution, while those outside this range are out-of-distribution. Throughout all datasets, we maintain uniform sampling across the support, allowing the support to uniquely characterize the data distribution.

### 2. Clarifying details regarding the CMA-ES Algorithm *[Addressing Reviewers ioAP and GKQk]*

- Constraint enforcement: While CMA-ES is an unconstrained optimization procedure, we ensure that misclassifications count as in-distribution adversarial attacks only if they satisfy both classification and distribution criteria. We have updated Algorithm 1 and the text in Sec.4 to clarify this.
- Added clarification to Figure 2c classification:
  - White points (InDist, correctly classified): $x_i \in [-10,10], y'_i=0$
  - Red points (In Dist failures): $x_i \in [-10,10], y'_i=1$
  - Black points (OOD): $x_i \not \in [-10,10]$
- Evaluation metric: We have added additional discussion regarding the metric to provide grounding for the readers, and guidelines for future benchmarks utilizing this metric.

### 3. Additional experiments with ImageNet *[Addressing Reviewers mN3d and GKQk]*

- Added comprehensive noise impact analysis:
  - Evaluated 27 distinct noise levels: three levels each for X, Y, and Z camera parameters
  - X-axis: horizontal translation
  - Y-axis: vertical translation
  - Z-axis: zoom effects
  - Found models particularly susceptible to Y-axis perturbations
  - Added Figure S6 showing detailed results across all conditions

### 4. Additional figure explaining CMA-ES *[Addressing Reviewer GKQk]*

- Added Figure S1 showing: Step-by-step visualization of CMA-Search algorithm, and Illustration of how constraints are enforced during search.
- Added examples of successful and unsuccessful search trajectories in the text.

### 5. Additional discussion on mitigating these attacks *[Addressing Reviewer GKQk]*

Added discussion of three promising mitigation strategies:
- Targeted Sampling near category boundaries where errors cluster
- Improved model initialization
- Multi-view Approach: Inspired by the saccadic movements of the human eye, a test-time ensemble of multiple viewpoints could be an interesting bio-inspired approach to mitigating these errors.

### 6. Additional discussion on the broader impact *[Addressing Reviewer GKQk]*

Added detailed discussion of security implications:
- Distinction from traditional adversarial attacks:
  - No engineered perturbations required
  - Exists naturally within expected data distribution
- Two critical vulnerabilities:
  - Bypasses traditional detection methods that rely on identifying out-of-distribution characteristics
  - Offers perfect plausible deniability to malicious actors
- Real-world implications:
  - Example: self-driving car crashes due to natural scene misclassification
  - Impossibility of determining deliberate vs accidental failures
  - Fundamental undermining of security approaches relying on tampering detection

### 7. References and Citations *[Addressing all Reviewers]*

- Fixed broken references
- Added citations to theoretical works, particularly: Gilmer et al. (2018), Fawzi et al. (2016, 2018).
- Added related work subsection on OOD detection and adversarial example detection.

We believe that these changes have comprehensively addressed the reviewers' questions, and significantly improved the evidence in support of our primary claim---"classification models, including object recognition models are deeply plagued by in-distribution adversarial attacks despite having near-perfect test set accuracies". Please let us know if any additional changes would be helpful.

---

### Author Response · Authors · 2024-12-18
**Follow-Up on Reviews and Decision Timeline**

Dear editorial team,

We deeply appreciate the time and the effort you and the reviewers have dedicated to evaluating our work.

As highlighted in our comments, we believe that our updated manuscript comprehensively addressed the reviewers' questions and provides strong evidence in support of our primary claim—classification models, including object recognition models, are deeply plagued by in-distribution adversarial attacks despite having near-perfect test set accuracies. Since the reviewers have not asked for any additional clarifications, we hope that our answers have sufficiently addressed their questions.

If there are any additional changes or clarifications that would be helpful in facilitating the decision-making process, please let us know. Thank you once again for your time and guidance.

---

### Decision · Action_Editor_tPzN · 2025-01-12

**Recommendation:** Accept as is

**Comment:**

The action editor agrees with the reviewers on acceptance. This study of in-distribution attacks, or perturbations in the hull of the training data, shows that deep networks for vision are susceptible to minor alterations to their inputs even in the absence of dataset bias. Controlled synthetic studies and experiments with standard natural image datasets expose and analyze this issue for transformations like viewing angle, rotation, relighting, and cropping. These are informative results that can spur further research into robustness beyond attacks in the form of engineered and optimized adversarial noise.

Reviewers raised a number of points that were resolved by the author responses and revisions. Initial concerns about the clarity of the definitions (ioAP, mN3d), exposition (GKQk), and discussion (GKQk) have all been settled. Further methodological detail and discussion about the optimization for finding the perturbations presented in this work (ioAP, GKQk) has been included in addition to futher results on ImagNet (mN3d, GKQk). The decision recommendations by reviewers confirm that their concerns have been addressed and no obstacle to acceptance remains.

The TMLR process has improved the work due to the detailed comments by the reviewers and the thorough revisions by the authors. The action editor thanks everyone for engaging and congratulates the authors on their publication in TMLR!

**Audience:**

All reviewers agree that there is an audience. The controlled and thorough experiments are certainly empirically informative, and these results are relevant for both the understanding of deep learning for vision and the security of deployed deep networks. The investigation of in vs. out of distribution data also provides material for the further study of generalization of deep networks with or without the adversarial element. Given the multiple and popular subjects touched on by this work, the action editor agrees that there is certainly an audience at TMLR.

**Claims And Evidence:**

All reviewers agree that the claims agree with the evidence. The definitions and conceptualization of in-distribution and out-of-distribution data are clear, adversarial and otherwise, and the scope of the analysis is in this work is adequate and expressed well. The empirical claims as to the types and degree of perturbations that are applicable and the types of networks that are susceptible are backed by experiments. The discussion is measured and not over-generalize or exaggerate the implications of the results.

This is an empirical work that is complementary to prior theoretical work. These theoretical works are properly credited and the contribution of this submission remains. The empirical bounds of the work are acknowledged, and where there are limits or approximations to the analysis (as for ImageNet) this is signposted.

The action editor finds no disagreement between claims and evidence to counter the consensus of the reviewers, and agrees on the connection of claims and evidence.

---

> ### Author Response · Authors · 2025-01-22
> **Camera ready submitted!**
>
> Dear Action Editor,
>
> Thank you for your kind words and for your thorough review of our work. We sincerely appreciate the constructive feedback and support from you and the reviewers, which have greatly enhanced the quality of our submission. We have now uploaded the camera-ready version as per the TMLR process! Please let us know if there are any additional steps we need to take!
>
> Warm wishes,
> Authors